# The Practical Price of Pyrrhonism

Jeremy Byrd 

Philosophy Department, Tarrant County College, Fort Worth, TX 76119, USA; jeremy.byrd@tccd.edu

**Abstract:** Sextus Empiricus presents Pyrrhonism as a skeptical lifestyle that is appealing, in large part, because of the tranquility it appears to afford. Addressing concerns about the practicality of such a lifestyle, Sextus suggests that Pyrrhonists can lead sufficiently ordinary lives while suspending belief about everything unclear. Here, I aim to offer a partial examination of the practicality and appeal of Pyrrhonism from the Pyrrhonist's perspective. In particular, I examine how a skeptic would likely respond if asked to consider his potential use of problematic concepts in his daily life. I argue that, even if the Pyrrhonist's skepticism is limited to certain types of controversial theoretical commitments, consideration of this issue would likely still cause him to worry that he is relying on beliefs about things unclear in his ordinary life. Along the way, I also hope to highlight some of the difficulties that a philosophically reflective person is likely to encounter if he is resistant to taking on philosophical commitments.

**Keywords:** skepticism; Pyrrhonism; Sextus Empiricus; ataraxia



## 1. Introduction

Assessing the practicality of Pyrrhonism in *Against the Logicians*, Sextus Empiricus tells us that, far from advocating for an approach that diverges sharply from ordinary life, Pyrrhonists "actually even speak on its side" [1] (8.158)[1]. While Sextus makes this claim in the context of a specific dispute with his philosophical rivals, it is also clear that, in general, he sees no difficulty in living a relatively ordinary life as a Pyrrhonist. *Prima facie*, this is difficult to accept. The Pyrrhonist that Sextus describes is an extreme skeptic who achieves peace of mind by suspending belief about everything unclear, treating all beliefs of this sort as ill-supported *dogmata*. While critics have raised several objections about the possibility and desirability of such a skeptical life, I wish to focus here on what perspective the skeptic should take on the practicality of a life devoid of such controversial beliefs.

In particular, I want to focus on how a Pyrrhonist would respond if asked to consider whether he makes use of problematic concepts in his daily life. I argue that, if he reflects on this issue, the skeptic is likely to suspend belief concerning the practicality of Pyrrhonism. Suspending judgment on this issue, however, means that the Pyrrhonist cannot be confident that he is not relying on false beliefs as he goes about his day. Consequently, if pressed on the details of what it takes to live an ordinary life while suspending judgment on everything unclear, the skeptic is likely to find himself subject to persistent and troublesome anxiety.

My argument is divided into five sections. I begin in the next section with a discussion of Sextus' explanation of how the skeptic can live by the appearances while suspending belief about everything unclear. The section that follows examines the type of relative tranquility that a skeptic could plausibly hope to achieve in this manner. For the Pyrrhonist, even such a relative peace of mind may hold considerable appeal when compared to the doubts and anxieties associated with dogmatism. In Section 4, I present a charitable interpretation of the extent of the Pyrrhonist's skepticism. This interpretation is guided by the goal of allowing the skeptic to maintain a wide range of beliefs, including some beliefs on controversial theoretical issues. I argue that such an interpretation offers the skeptic the best opportunity to live a relatively ordinary life. In this section, I also contend

that, despite his skepticism, the Pyrrhonist is likely to remain philosophically reflective and open to continuing his theoretical investigations. In Section 5, I discuss why the practicality of Pyrrhonism will depend on whether the skeptic's beliefs make use of concepts that require commitments about things unclear, and I argue that reflecting on this issue would likely lead the skeptic to suspend belief. Finally, in Section 6, I consider and reply to some potential weaknesses in my argument.

I conclude that, faced with the objection that an ordinary life requires commitments on things unclear, the skeptic would likely find that he is unable to escape the worry that Pyrrhonism is impractical. In making this case, I also hope to highlight some of the difficulties that a philosophically reflective person is likely to encounter if he also seeks to resist controversial philosophical commitments. Nevertheless, such skepticism might still hold some attraction for those suitably inclined. At least, I will not argue otherwise here. And so, for present purposes, I will concede that the skeptic may continue to find Pyrrhonism appealing, so long as the goal is only to achieve relative tranquility.

## 2. Living by the Appearances

Pyrrhonism is distinguished from other versions of skepticism by an emphasis on *epochē*, or suspension of belief about everything unclear. Sextus describes Pyrrhonists as suspending belief on a wide range of topics, given the equal force (*isostheneia*) of the evidence for and against the relevant positions. There is considerable controversy concerning how wide this range is. In the *Outlines of Pyrrhonism*, Sextus tells us that the Pyrrhonist does not accept any position on matters that are unclear, including the truth of the skeptical claims he might appear to make [2] (1.14–15). Thus, a Pyrrhonist may seem to make choices and live his life without a substantial number of the beliefs that most of us rely upon in our daily lives. It is unsurprising that, to many of Sextus' critics, a life of this sort appears immensely impractical, if not bizarre.

Yet Sextus assures us that, because Pyrrhonists "attend to what is apparent" [2] (1.21) and "live in accordance with everyday observances, without holding opinions" [2] (1.23), their lives are at least superficially ordinary. Sextus offers this assurance in reply to the charge that we cannot act without belief, and he observes that to live by the appearances (*phantasiai*) is to rely on "guidance by nature, necessitation by feelings, handing down of laws and customs, and teaching of kinds of expertise" [2] (1.23). In the lines that follow, he explains that this amounts to relying on the way that we tend to perceive and think about the world around us, responding to natural feelings such as hunger and thirst, behaving in accord with social conventions, and using the skills acquired in our studies. In these respects, the Pyrrhonist is supposedly much the same as the ordinary people around him, except that while he is relying on these perceptions, thoughts, feelings, conventions, and skills, he does not form any beliefs about anything unclear [2] (1.13), does not make judgments about whether addressing these feelings or following these conventions is truly good [3] (11.158 and 166), and merely follows "what seems to be expedient" as he practices his profession [2] (1.237).

Sextus here extends appearances to include not only the way the world around us appears to us through perception, but also the way that a claim can appear to be true and an argument can appear to be sound. This broad sense of appearance may help him avoid the charge that the Pyrrhonist must form beliefs about the *isostheneia* of the arguments that lead him to suspend judgment. The Pyrrhonist need not believe that the arguments are equally compelling. Instead, being guided by the *phantasiai*, he suspends judgment as a response to the fact that they appear so to him[2].

Even if we grant Sextus that the skeptic can be guided by the apparent strength of the arguments, many are still concerned that suspending belief on everything unclear would leave the Pyrrhonist incapable of action. This is the well-known *apraxia* objection. Of course, sometimes, we all suspend judgment on difficult issues when we feel the available evidence is not decisive. In suspending judgment about everything unclear, however, the Pyrrhonist goes well beyond any ordinary uncertainty we may feel about such matters. Yet the skeptic

must still act, and for many it is difficult to see how he could do so without relying on some controversial beliefs.

Thus, Hume famously dismisses Pyrrhonism in *An Enquiry Concerning Human Understanding*, contending that "[n]ature is always too strong for principle" [6] (12.23). The worry is that so long as the Pyrrhonist does not accept any controversial beliefs, he will spend his time dawdling about, uncertain of what to do. Consider, for example, the role of value judgments in deliberation. There are many debates about value, and Sextus tells us in the *Outlines* that the skeptic suspends belief about whether anything is actually good or bad [2] (1.27–28 and 3.235–238)[3]. If a Pyrrhonist does not make value judgments, however, then how would he make up his mind about what to do? Indeed, Pyrrho himself is often anecdotally caricatured as so lacking in commonsense as to display a bizarre disregard for his own welfare[4]. If that is the practical consequence of Pyrrhonism, then it is hard to disagree with Hume about the impossibility of such a thoroughly skeptical life.

In *Against the Ethicists*, Sextus considers this worry that Pyrrhonism inevitably results in either inactivity or inconsistency:

> . . . to inactivity, because, since the whole of life is bound up with choices and avoidances, the person who neither chooses nor avoids anything in effect renounces life and stays fixed like some vegetable, and to inconsistency, because if he comes under the power of a tyrant and is compelled to do some unspeakable deed, either he will not endure what has been commanded, but will choose a voluntary death, or to avoid torture he will do what has been ordered, and thus no longer 'Will be empty of avoidance and choice', to quote Timon, but will choose one thing and shrink from the other, which is characteristic of those who have apprehended with confidence that there is something to be avoided and to be chosen. [3] (11.163–164)

Sextus, however, sees no compelling reason to accept this dilemma, telling us instead that the Pyrrhonist may choose based on "the preconception (*prolēpsei*), which accords with his ancestral laws and customs" [3] (11.166). The Pyrrhonist will do what appears best to him, influenced by his upbringing and the customs of his community, while still suspending judgment on everything unclear.

## 3. Achieving Peace of Mind

According to Sextus, the apparent reward for learning to live without beliefs about anything unclear is peace of mind (*ataraxia*). In the *Outlines*, he tells us that the road to Pyrrhonism starts when the future skeptic is troubled by the realization that there is a tremendous amount of uncertainty on a range of issues of concern to him, motivating him to investigate to discover the truth and achieve peace of mind [2] (1.26–29). Unfortunately, pursuing these investigations, he grows less confident in the answers he finds. On issue after issue, he wavers in his commitments as he confronts the apparent *isostheneia* of the evidence. Eventually, frustrated by the failure of his efforts to discover the truth, he reacts by acknowledging the depth of his ignorance and suspends judgment.

In this moment of defeat, however, Sextus reports that the Pyrrhonist surprisingly finds the peace of mind he has been seeking all along. In a well-known passage, Sextus compares this to Apelles' discovery of an artistic technique that eluded him until he gave up in frustration:

> They say that he was painting a horse and wanted to represent in his picture the lather on the horse's mouth; but he was so unsuccessful that he gave up, took the sponge on which he had been wiping off the colours from his brush, and flung it at the picture. And when it hit the picture, it produced a representation of the horse's lather. Now the Sceptics were hoping to acquire tranquility by deciding the anomalies in what appears and is thought of, and being unable to do this they suspended judgement. But when they suspended judgement, tranquility followed as it were fortuitously, as a shadow follows a body. [2] (1.28–29)

Pyrrhonism is thus presented, at least at the start, as a psychological response to the frustration of failing to discover the truth about the world[5]. Vexed by his uncertainty, the incipient Pyrrhonist is driven to pursue the truth in order to ease his troubled mind, only to discover that, when he finally accepts that he does not know what to believe, he suddenly stumbles upon the tranquility he had been pursuing. The Pyrrhonist has not found the truths he seeks, but by a happy accident he does find peace of mind.

What is the connection between *epochē* and *ataraxia?* Returning to the tyrant example, Sextus suggests that the skeptic would endure such a demanding situation better than most, regardless of which choice his upbringing leads him to favor, precisely because he lacks the dogmatist's additional beliefs about what is truly good or bad [3] (11.166)[6]. Likewise, in other places in both the *Outlines* and *Against the Ethicists*, Sextus suggests that the Pyrrhonist achieves peace of mind by suspending certain types of value judgments.

For example, in the *Outlines*, he indicates that the source of mental disturbance is the belief that some things are good or bad by nature:

> For those who hold that things are good or bad by nature are perpetually troubled. When they lack what they believe to be good, they take themselves to be persecuted by natural evils and they pursue what (so they think) is good. And when they have acquired these things, they experience more troubles; for they are elated beyond reason and measure, and in fear of change they do anything so as not to lose what they believe to be good. But those who make no determination about what is good and bad by nature neither avoid nor pursue anything with intensity; and hence they are tranquil. [2] (1.27–28)

In this passage, the Pyrrhonist's tranquility is a result of suspending his beliefs about what is good and bad by nature, not suspending judgment in general. Sextus makes a similar point in *Against the Ethicists*, where he boldly claims that "[a]ll unhappiness, therefore, comes about by way of the pursuit of good things as good and the avoidance of bad things as bad" [3] (11.113)[7]. While the Pyrrhonist is not able to ignore or remain untroubled by naturally unpleasant sensations, such as hunger and thirst (11.148–149), he is supposedly able to remove the additional worries that come from having strong convictions about what is truly good or bad. Lacking these additional concerns, the skeptic achieves a relative tranquility.

On this reading, while the Pyrrhonist is not completely undisturbed, he is comparatively tranquil insofar as he lacks the anxiety that others feel because of their judgments about what is truly good or bad. Even so, the Pyrrhonist is still able to be guided by his upbringing and the customs of his community. Given what he was taught about morality, certain actions are still likely to appear good or bad to him, and he is still likely to care about the outcomes of his actions as a result. Accordingly, despite his skepticism, the Pyrrhonist is not emotionally disengaged. Rather, he is supposedly less disturbed by what happens because his feelings are more moderate. As far as things appear good or bad to him, he naturally cares about the decisions he makes and the consequences of his actions. Yet his Pyrrhonism supposedly eases these concerns significantly and leaves him less troubled than those of us who are concerned to obtain and keep what is truly good and avoid what is truly bad. And so, while the skeptic still avoids what appears bad and pursues what appears good, he does not do so with intensity.

While I find this reading of Sextus on *ataraxia* plausible, I will not defend it here[8]. Nor will I discuss the apparent plausibility of achieving such comparative tranquility either by suspending belief in everything unclear generally or by suspending certain types of value judgments in particular[9]. It is enough for our purposes that such relative peace of mind is a far more realistic goal for the skeptic. There is no skeptical route to complete tranquility. Even a Pyrrhonist would be bothered by the threat of torture. Likewise, as I argue below, a skeptic would likely be troubled if it appeared to him that he might still have false beliefs about things unclear. Nevertheless, assuming that beliefs about what is truly good or bad do add to our worries, pains and doubts would be less intensely troubling for the skeptic, since he does not believe that these things are truly bad[10].

## 4. Suspending Judgment

How far does the Pyrrhonist go in the pursuit of tranquility? In several passages, Sextus suggests that the Pyrrhonist suspends all belief. For example, in the *Outlines*, he remarks:

> The chief constitutive principle of scepticism is the claim that to every account an equal account is opposed; for it is from this, we think, that we come to hold no beliefs (*mē dogmatizein*). [2] (1.12)

Yet, in the much-discussed passage that immediately follows, he offers a significant qualification:

> When we say that Sceptics do not hold beliefs (*mē dogmatizein*), we do not take 'belief' (*dogmatos*) in the sense in which some say, quite generally, that belief is acquiescing in something; for Sceptics assent to the feelings forced upon them by appearances—for example, they would not say, when heated or chilled, 'I think I am not heated (or: chilled)'. Rather, we say they do not hold beliefs (*mē dogmatizein*) in the sense in which some say that belief (*dogma*) is assent to some unclear object of investigation in the sciences; for Pyrrhonists do not assent to anything unclear. [2] (1.13)

Likewise, Sextus also assures us that the skeptic will not deny that it appears to him that honey sweetens when he has the sensation of its sweetness [2] (1.20). But when the Pyrrhonist assents to such *phantasiai*, does this mean that he forms a belief about the warmth of his surroundings and the sweetness of the honey?

Fortunately, we do not need to settle the debate on this issue here[11]. Instead, I will assume that many ordinary beliefs, such as the belief that honey is sweet, do not fall within the scope of the skeptic's *epochē*[12]. On this interpretation, when skeptics assent to the *phantasiai* that are forced upon them, they form beliefs about the world around them that help guide them in their daily lives, but they remain unpersuaded by any argument about the underlying nature of things. Suspension of belief is reserved for theoretical commitments on controversial issues, and only as far as such a belief would be the result of being persuaded by argumentation. These are the *dogmata* that fall within the range of the skeptic's *epochē*. The Pyrrhonist believes that honey is sweet but is unconvinced by any argument to accept *dogmata* about the true nature of either honey or sweetness.

This reading allows the Pyrrhonist to hold beliefs on several contested issues. Recall that, according to Sextus, appearances are influenced by both culture and education, in addition to human nature and experience. Thus, if assenting to the *phantasiai* allows the Pyrrhonist to believe that things are as they appear to be, then his skepticism does not prevent him from forming some controversial beliefs. Raised within a polytheistic culture, for instance, the Pyrrhonist can "accept, from an everyday point of view, that piety is good and impiety bad" [2] (1.24) and that there are gods he should worship and revere [2] (3.2). Such skepticism is compatible with beliefs on issues as controversial as value and religion, so long as the skeptic is not convinced by the arguments in favor of his beliefs[13]. Such everyday beliefs do not count as *dogmata*.

While I again find this interpretation quite plausible on its own, the main advantage of limiting the scope of the Pyrrhonist's *epochē* in this way here is that it offers the best chance for a skeptic to live an ordinary life. A Pyrrhonist cannot be rationally persuaded to accept some controversial theoretical commitment and remain a Pyrrhonist. Yet this interpretation allows that the skeptic can share many of the same beliefs as those around him, even when such beliefs involve assenting to the appearances on contested issues such as the value of piety or the existence of the gods. Accordingly, we are allowing the skeptic the maximum range of beliefs that might still be consistent with a plausible notion of avoiding assent to everything unclear. If the Pyrrhonist still finds the skeptical lifestyle impractical, a more expansive *epochē* would not help.

Still, while the skeptic is unpersuaded by any argument to take on theoretical commitments, this does not mean that the Pyrrhonist has no interest in theoretical investigation, as

some claim[14]. Instead, the Pyrrhonist continues to investigate and seek the truth about the underlying nature of things, and he does so with an open mind because he does not believe that he has already found the answer and because he also does not believe that answers to such questions are impossible to find. Indeed, there is considerable evidence supporting the skeptic's interest in continuing his philosophical investigations[15].

Sextus tells us at the outset of the *Outlines* that skeptics remain active inquirers:

> When people are investigating any subject, the likely result is either a discovery, or a denial of a discovery and a confession of inapprehensibility, or else a continuation of the investigation. This, no doubt, is why in the case of philosophical investigations, too, some have said that they have discovered the truth, some have asserted that it cannot be apprehended, and others are still investigating. Those who are called Dogmatists in the proper sense of the word think that they have discovered the truth—for example, the schools of Aristotle and Epicurus and the Stoics, and some others. The schools of Clitomachus and Carneades, and other Academics, have asserted that things cannot be apprehended. And the Sceptics are still investigating. [2] (1.1–3)

Here, a contrast is drawn between the dogmatist, who thinks he does not need to continue his investigations because he already has discovered the truth, and the Academic, who supposedly sees no point in continuing his investigation because the truth cannot be found. The Pyrrhonist is distinguished from both because he does not believe he has discovered the answers to his philosophical questions so far and he remains open to the possibility that such answers might yet be found. And so, his philosophical investigations continue[16].

Given that the Pyrrhonist remains open to the possibility that he may yet be able to find the philosophical answers he seeks, he is not likely to give up the search unless he loses his interest in discovering these truths. Recall, however, that the initial pathway to Pyrrhonism, according to Sextus, started from a concern with finding philosophical truths. The frustration of failing to find the answers he seeks may prompt the skeptic to suspend belief about dogmatic claims, but it also indicates just how much he valued the truth.

Moreover, the Pyrrhonist may even believe, at least from an "everyday point of view", that philosophical truths are worth pursuing. As Dan Moller observes, there are many philosophical issues where none of the relevant positions are likely to appear true to the skeptic once he has suspended belief [14] (pp. 433-435). Yet, we also cannot simply assume that no appearances on controversial topics survive the skeptic's *epochē*, since some clearly do. After all, *ataraxia* still appears valuable to Sextus, despite his skepticism[17]. Likewise, we are allowing that the skeptic can accept the apparent value of piety and believe in the gods. So why do some appearances survive while others do not?

Moller offers a plausible conjecture. Remember that, in reply to the *apraxia* objection, Sextus tells us that, like everybody else, the skeptic is affected by his nature, his environment, and his experiences. Moller's suggestion is that *phantasiai* will tend to persist for the skeptic even after he suspends belief when they are the result of one of these influences. So, hunger still hurts the skeptic because this is a natural sensation, and it still appears bad because we have a natural aversion to it. Likewise, it can still appear appropriate to the skeptic to be pious because the value of piety has been central to the cultural and religious tradition in which he has been immersed.

Accepting Moller's conjecture, let us say that the skeptic can accept that piety is good after growing up in a religious culture that emphasizes this value. If so, then he is also likely to accept the apparent value of discovering the answers to fundamental questions about the nature of reality. Valuing such truths may be quite natural, much like the aversion to hunger, or it may at least be natural for anybody drawn to philosophical investigation in the first place[18]. If not, it is certainly something that is likely to be highly valued in a community that would foster the type of philosophical curiosity needed to provide a path to Pyrrhonism, a value that has been passed on to the skeptic. And so, given the skeptic's openness to discovering the answers to philosophical questions and likely interest in finding

those answers, I suggest we take Sextus at his word when he tells us that skeptics have not disengaged from philosophical inquiry.

## 5. Believing from an Everyday Point of View

When Sextus tells us that the skeptic can believe in the value of piety and in the gods from an everyday point of view, these everyday beliefs are only compatible with skepticism so long as they do not require any dogmatic commitments. Likewise, when the skeptic accepts that he is cold or that honey is sweet, such ordinary beliefs would be incompatible with skepticism if they require taking a stand on anything unclear. Are beliefs such as these insulated, however, from theoretical commitments that would prove unacceptable to the Pyrrhonist[19]? More importantly for our present concerns, can a skeptic answer this question without accepting some *dogma*?

To appreciate why everyday beliefs might be thought to require dogmatic commitments, let us focus on the concepts at work in the skeptic's beliefs. When a skeptic assents to an appearance, the concepts he uses must be distinct from the dogmatic concepts that Sextus criticizes as apparently incoherent or inapprehensible. Or rather, if the skeptic reflects on his concepts, he will be disturbed if they do not at least appear nondogmatic to him. For instance, after a lengthy account of all the several types of motion that are fundamental to natural science in the *Outlines*, Sextus concludes: "This will be enough, in an outline, about the kinds of motion; and it follows that the natural science of the Dogmatists is unreal and inconceivable (*anepinoēton*)" [2] (3.114). This claim, that some extraordinarily important concepts involved in the dogmatists' beliefs about the world appear to be inconceivable or inapprehensible (*akatalēptos*), is not unusual for Sextus. Elsewhere in the *Outlines*, he makes similar comments about dogmatic concepts of human beings (2.22 and 29–31), appearances (2.70–71), signs (2.104 and 110), gods (3.5), and causes (3.13 and 23).

If the dogmatists' concepts appear incoherent or inapprehensible, then presumably the skeptic would be disturbed if it seems to him that he is using them. If the skeptic can make no sense of the dogmatists' concepts of motion, then he would be troubled if he appears to himself to believe something is in motion in a dogmatic sense. Thus, if a skeptic reflects on his acceptance of the apparent motion of a ship, he can only maintain his *ataraxia* if his belief appears to him to make use of some distinct conception of motion, a conception free from whatever problems appear to threaten its dogmatic counterparts. Likewise, since the skeptic is prepared to accept *phantasiai* about humans, appearances, signs, gods, and causes, the untroubled skeptic who reflects on his concepts must take himself to be using nondogmatic notions of each of these, which appear neither incoherent nor inapprehensible.

Perhaps the concepts at work in the skeptic's everyday beliefs are just ordinary concepts. Benson Mates considers this as a way to make sense of how the Pyrrhonist can assent to the appearances while still suspending judgment about everything unclear:

> One proposed solution is this. It is suggested that Sextus, representing the Pyrrhonists, in effect distinguishes two senses or uses for a sentence like "The honey is sweet." One of these is its "ordinary" use, which it has in the discourse of daily life; the other is a "philosophical" use, according to which it means something like "The honey *really* is sweet" and according to which it is loaded with philosophical presuppositions about what honey really is and about how we acquire knowledge of an external world. [17] (p. 28, emphasis in original)

Thus, according to this proposal, the skeptic can share many of our ordinary beliefs despite his philosophical skepticism, because ordinary concepts and beliefs involve no suspect philosophical commitments.

The problem is that it seems dogmatic to claim that ordinary concepts do not come with dogmatic commitments. Consider, for example, the Platonic view, according to which many of our most important ordinary concepts are reflections of the forms. Such a theory would suggest that our ordinary concepts carry implicit philosophical commitments. Of course, in *Against the Logicians*, Sextus criticizes the Platonic theory of concepts [1] (8.56–62), along with the Epicurean alternative (8.337). In context, however, such criticisms are

meant to help skeptics suspend belief about the truth of these dogmatic theories, not as conclusive grounds for rejecting the Platonic approach in favor of a skeptical theory that leaves ordinary concepts insulated from philosophical disputes. Further, there is no reason to suppose that ordinary concepts are likely to appear insulated to the Pyrrhonist because of his nature or any of the other influences that Sextus lists. So why would a skeptic not treat such a theory of ordinary concepts as just another dogmatic alternative to the views of Plato and Epicurus?

In fact, many of our ordinary concepts seem to include or commit us to beliefs that the skeptic would find philosophically suspect. For example, consider our ordinary concepts of composite objects. In *Against the Physicists*, Sextus argues against dogmatic mereological views. On the one hand, he tells us that both "plain experience" and the dogmatist's concept of a whole seem to show that a whole cannot be separate from its parts [4] (9.338–340). On the other hand, if the whole is not separate from the parts, then it seems there really is no whole after all. Just as there is no "roofing apart from beams arranged in a certain way," there is no whole but only a particular arrangement of smaller objects (9.338–340). Regardless of how persuasive we find this argument, Sextus' attack on the mereological views of philosophers would seem to apply equally well (or poorly) to our ordinary concepts of wholes and parts.

If these ordinary concepts are not equally suspect to the skeptic who reflects on these issues, then they must appear immune from such doubts. Once again, however, it is hard to see why a Pyrrhonist should not treat a belief in their immunity as just another belief about things unclear. Moreover, even if ordinary concepts and beliefs are, in fact, neutral on all philosophical issues, any argument about the philosophical commitments of our ordinary beliefs is still likely to strike the skeptic as dogmatic. Whatever the truth may be about the philosophical implications of ordinary concepts and beliefs, a Pyrrhonist who reflects on the philosophical disputes on this issue is likely to find the arguments on each side equally compelling and suspend judgment.

To avoid any dogmatic claims about ordinary concepts and beliefs, a Pyrrhonist might instead turn to a suggestion by Bredo C. Johnsen [5] (pp. 554–560)[20]. Explaining how we might distinguish skeptical and dogmatic concepts, Johnsen follows Mates and proposes that the former do not draw on nor align with any dogmatic philosophical theory. In thinking about honey, for example, the skeptic would conceive of honey as "*the sort of thing—if such there be—that appears (has appeared) to be thus and so*, or as the sort that normally appears thus and so" (p. 557, emphasis in original). Employing such a skeptical concept, the skeptic could report that honey tastes sweet without fear of incoherence or incomprehensibility, and likewise for claims about motion, humans, appearances, etc.

What distinguishes Johnsen's proposal is that such skeptical concepts are also supposed to be distinct from our ordinary concepts as well, precisely because of the worry that ordinary concepts carry commitments that the skeptic would not accept. Thus, according to Johnsen, the ordinary concept of honey "clearly includes its *being* sweet, independently of its appearing to anyone in particular to be so" (p. 558, emphasis in original). If Johnsen is right, then our ordinary concept of honey commits us to a belief about the nature of honey. In contrast, the skeptic supposedly avoids any belief about the true nature of honey, while still believing that honey is sweet.

By now, the problem with this proposal from the skeptic's perspective should be clear. Perhaps the skeptic does make use of concepts that are free from dogmatic commitments. Perhaps these concepts are also distinct from our ordinary concepts, which are not insulated in this way. Yet any theory that says so would likely seem dogmatic to a Pyrrhonist, as he is unlikely to have any *phantasia* about the types of concepts that can be employed in beliefs or about whether any of them can be used without commitment to *dogmata*.

Overall, we should not be surprised that skeptics would not accept any theory about their own concepts. As Richard Bett observes, a Pyrrhonist "does not adopt any philosophical theories, whether about the nature of concepts or about anything else" [28]. No theory about the nature of the concepts at work in the skeptic's everyday beliefs would

likely appear true to the Pyrrhonist. Neither nature nor any other candidate influence would likely support such *phantasia*. Consequently, if asked to consider the nature of his concepts, the skeptic would face considerable uncertainty about whether it is possible to live an ordinary life without relying on potentially false *dogmata*.

## 6. Lingering Doubts

This argument assumes, however, that the skeptic still uses such potentially worrisome concepts. Earlier, I suggested that a charitable interpretation would allow the skeptic a considerable number of beliefs about the world, as this would allow him the best opportunity to live a sufficiently ordinary life. These beliefs have intentional content. They are about the world. Moreover, this content is conceptual[21]. If the skeptic believes that honey is sweet, he does so by applying a concept of honey and a concept of sweetness. If he believes a ship is in motion, he does so by applying a concept of ships and a concept of motion. Some might be suspicious of my charity, however, and wonder if the skeptic could avoid the worry that he is using problematic concepts by avoiding beliefs altogether and relying merely on the appearances instead[22].

To make sense of how the skeptic can rely on the appearances in this way, we must make two assumptions. First, *phantasia* would need to be distinct from beliefs yet have intentional content that can help guide the skeptic through his daily life. Second, while these appearances are about the world, their intentional content would need to be nonconceptual. Let us take up each of these considerations in turn.

For the skeptic to live by the appearances without beliefs, his *phantasiai* would need to be roughly functionally equivalent to the ordinary beliefs that typically guide our decisions and actions. Here, for instance, is how Casey Perin, adopting such an interpretation, summarizes Sextus' reply to the *apraxia* objection:

> The Sceptic has no beliefs about how things are, but he has (and cannot avoid having) appearances, and these appearances, together with his desires, are sufficient for action. The explanation of any action the Sceptic performs will have the form '*S* does action *A* because he desires to $\varphi$ and it appears to him that *p*' where *p* is, typically, a complex proposition that relates the doing of *A* to $\varphi$-ing and so to the satisfaction of the desire to $\varphi$. For example, the Sceptic drinks a glass of water because he is thirsty, i.e., has a desire to drink, and it appears to the Sceptic that there is a glass of water in front of him and that he can drink it. The appearance to which the explanation of the Sceptic's action appeals is not a belief, but it is an analogue to belief in the sense that it plays the role in the explanation of the Sceptic's actions that belief plays in the explanation of the non-Sceptic's actions. [9] (pp. 96–97)

While Perin seems to assume that appearances have conceptual content, we can set this aside for the moment. Still, if the skeptic is to live without beliefs, his *phantasiai* would need to be psychological states that function in much the same way that beliefs do, so much so that the skeptic can live an otherwise ordinary life without ordinary beliefs about the world around him[23].

Unsurprisingly, the view that there are such distinct but functionally similar psychological states is controversial. Galen, for instance, rejects the claim that a skeptic can live an ordinary life without ordinary beliefs, in part because he seems to hold that any psychological state that is relevantly functionally equivalent is just a belief by another name[24]. In particular, he contends that a skeptic cannot lack ordinary beliefs without some observable and significant difference in his behavior. Indeed, Galen contends that, so long as the skeptic can make choices and act, this is enough evidence that he has ordinary beliefs.

Of course, Galen could be wrong. Julia Annas and Jonathan Barnes argue instead that, when it comes to values, there is a real difference between being guided by the *phantasiai* and believing the world is as it appears [29] (p. 169). If, because of his upbringing, it appears to the skeptic that piety is good, then he will tend to refrain from impiety in a way that is

externally unremarkable. Yet, according to Annas and Barnes, such a skeptic is remarkably different from the rest of us on the inside, precisely because he does not believe that piety is really good.

Once again, we do not need to settle this debate. What is significant for our present concerns is that there is a debate, and in response to such a debate the skeptic would likely suspend belief on this issue. When different philosophers provide arguments for and against a view, Sextus often suggests that this provides enough motivation for the skeptic to suspend belief[25]. For instance, in his discussion in the *Outlines* of the various types of considerations that lead the Pyrrhonist to suspend judgment, Sextus tells us that, for the skeptic, consideration of such differences of opinion results in *epochē*:

> For we shall be convinced either by all humans or by some. If by all, we shall be attempting the impossible and accepting opposed views. But if by some, then let them say to whom we should assent. The Platonist will say 'to Plato', the Epicurean 'to Epicurus', and the others analogously, and so by their undecidable dissensions they will bring us round again to suspension of judgement. [2] (1.88)

Thus, it seems that, if the skeptic considers the controversy over whether he can live an ordinary life without ordinary beliefs, he would likely suspend judgment.

Nor is there any reason to think the skeptic is likely to have higher-order *phantasia* about the distinction between beliefs and appearances. Claims about *phantasiai* are philosophically sophisticated and controversial, and it seems highly doubtful that nature or nurture would encourage any view on them. Consequently, if asked to reflect on this debate, the skeptic would likely be left unsure whether he possesses ordinary beliefs.

Much the same can be said about the assumption that the skeptic's *phantasiai* have non-conceptual content. There is an ongoing controversy about whether certain psychological states, such as perceptions, have such content. For instance, our perceptions of the world may be too rich and fine grained to be conceptual in nature. Perhaps such considerations suggest that, for the skeptic, the apparent sweetness of the honey or the apparent motion of a ship is nonconceptual.

It strikes me as less plausible that all nonperceptual *phantasia* have nonconceptual content. For present purposes, however, let us assume for the moment that, when it appears to the skeptic that piety is good or that the gods exist, the intentional content of such appearances is nonconceptual as well. Even so, the familiar problem remains. Given the predictable controversy on these issues, a skeptic who reflects on these debates would likely suspend judgment on the intentional content of appearances, and there is no reason to think that his *phantasiai* would likely appear nonconceptual to him once he does so. If left unsure whether his appearances make use of problematic concepts, however, then he would face the same worry about whether it is possible to live by the appearances without relying on potentially false dogmata.

Worries such as these only arise, however, if the skeptic bothers to reflect on his potential use of problematic concepts. If he never considers whether his beliefs or *phantasia* make use of concepts that require dogmatic commitments, then this issue need never bother him. If Annas and Barnes are right, his internal life may still differ in significant ways from ours. Likewise, if the rest of us rely on beliefs about things unclear, then he may make his choices and think about the world much differently than we do. Nevertheless, he can still watch the ships moving at sea, and he can still go to the market to buy some honey. Moreover, he can go about his day while still occasionally engaging in philosophical reflection and investigation, so long as these reflections and investigations never lead him to consider the doubts raised here.

Once his use of concepts is raised as an issue, however, the reflective skeptic should be open to investigating whether he makes use of problematic concepts in his daily life. It is in this moment of reflection on how he can live by the appearances that the Pyrrhonist would experience a worry that his skepticism cannot address. While he is open to discovering the truth about these concepts, we can expect that the skeptic would suspend judgment instead. Sextus, of course, tells us that suspending belief about everything unclear as how

the skeptic first achieved *ataraxia*. In this case, however, the skeptic is unlikely to find relief from the worries that disturb his peace of mind. Regardless of whether he relies on beliefs or *phantasia* as he goes about his day, suspending belief on these issues would mean he cannot rule out the possibility that he is using concepts that require dogmatic commitments about things unclear.

Compare the Pyrrhonist's position here to that of philosophers who reject the claim that ordinary concepts carry any significant philosophical commitments[26]. These philosophers are free to question or even reject philosophical claims that might superficially seem to put them in opposition to our ordinary beliefs. For example, philosophers who reject the existence of composite objects also probably talk about tables and chairs in their ordinary lives. They are not likely to find this troubling, however, so long as they also hold that we can use our ordinary concepts of tables and chairs without taking a stand on controversial mereological issues. Suspending belief on whether ordinary concepts can be used in this way puts the Pyrrhonist at a comparative disadvantage.

If the skeptic is fortunate, he may go about his daily life and engage in philosophical discussions without ever considering these issues. After all, while Sextus shows a clear interest in examining both concepts and philosophical theories of concepts, he never seems to consider worries such as these about the skeptic's concepts. There are, however, lively and ongoing philosophical debates about whether ordinary concepts carry controversial commitments, and in the long run, such ignorant tranquility is not likely to endure. Once the skeptic turns his attention to the problem of whether a practical Pyrrhonism requires the use of problematic concepts, his skepticism would leave him poorly positioned to address it.

## 7. Conclusions

While a philosophically reflective Pyrrhonist would presumably be troubled by the prospect that he continues to rely upon beliefs about things unclear, I do not mean to suggest that a Pyrrhonist could not live relatively well in his daily life by trusting the appearances. Nor do I mean to suggest that the worries raised here would be pressing on him throughout his day, even if he is aware of them. Like most of us, the skeptic is likely untroubled by such philosophical concerns in much of his everyday life. There is no reason to suspect worries about his concepts would weigh upon him while he is at the market or during any of his other routine activities. If pressed to consider, however, whether his daily life requires the use of concepts carrying dogmatic commitments, the skeptic would likely suspend judgment about whether he is still relying on beliefs about things unclear.

Doubts about the practical limits of his skepticism would likely prove troubling for the skeptic, if either his nature or his culture leave him concerned to avoid false beliefs. Nevertheless, if a significant part of the appeal of Pyrrhonism lies in its apparent ability to provide the skeptic with at least a relative peace of mind, then perhaps the skeptical life may still appear worth pursuing. I have my doubts about the value of such a limited *ataraxia*[27]. Likewise, I am unconvinced that suspending judgment about everything unclear is likely to lead the skeptic to even this relative peace of mind[28]. Nevertheless, given the troubles he is liable to face as he tries to live an ordinary life without so many of the beliefs on which most of us rely, such a limited *ataraxia* is the most the Pyrrhonist could plausibly hope to achieve.

**Funding:** This research received no external funding.

**Institutional Review Board Statement:** Not applicable.

**Informed Consent Statement:** Not applicable.

**Data Availability Statement:** Not applicable.

**Conflicts of Interest:** The author declares no conflict of interest.

## Notes

1　　Some of Sextus' works were once mistakenly thought to be part of his *Against the Mathematicians* (*M*), and it is still customary to cite them as such. Accordingly, the two books of *Against the Logicians* are cited as *M* 7 and 8. Likewise, *Against the Ethicists* is cited as *M* 11. The two books of *Against the Physicists* are cited as *M* 9 and 10. For ease of exposition, I will often follow Sextus' lead from the *Outlines of Pyrrhonism* [2] (1.14–15) and write as if he is making claims about Pyrrhonism, while reminding the reader at the outset that Sextus may not be committed to the truth of such claims. Translations are taken from [1–4].

2　　See [5]. For present purposes, I am also willing to allow that the skeptic can believe that the arguments on each side of an argument are equally compelling, so long as he forms this belief based on the appearance and is not persuaded by an argument for the *isostheneia* of the evidence.

3　　Surprisingly, in *Against the Ethicists*, Sextus appears to suggest that Pyrrhonists believe that nothing is actually good or bad. See [3] (11.68–95). For a plausible interpretation of this discrepancy, see [7] (pp. 207–215).

4　　For an insightful discussion of these anecdotes and their veracity, see [7] (pp. 63–111).

5　　While some philosophers might hold that suspending belief is the only rational response to being confronted with equally compelling evidence for and against some proposition, the Pyrrhonist would presumably also suspend belief about the requirements of rationality. If his frustration leads him to suspend belief, however, then there is no need for the skeptic to accept such a potentially controversial epistemic principle. On this point, see also [5] (p. 529) and [8] (pp.15–16). Nevertheless, it is at least arguable that it would still appear rationally appropriate to the skeptic to suspend judgment given the apparent *isostheneia* of the evidence on both sides of some issue. See [9] (pp. 33–58). While I will not explore this issue further here, I have my doubts about whether *phantasiai* about the requirements of rationality would be likely to remain once the Pyrrhonist considers the relevant arguments on each side. The basis for my doubts should hopefully become clear below.

6　　Richard Bett argues that it is unlikely that a Pyrrhonist would be willing to endure significant pain and hardship in order to follow convention [10] (p. 11). Since Sextus concedes that Pyrrhonists will still share an aversion to pain, Bett claims that it is implausible that a Pyrrhonist would be willing to endure torture based on the influence of conventional values if he is not intensely committed to these values.

7　　There is a tension between the claim that tranquility comes from suspending judgment in general and the claim that the Pyrrhonist achieves tranquility through suspending certain value judgments, as Bett observes [10] (p. 9). See also [11].

8　　For helpful discussions of this reading, see [12] (pp. 322–324), [13] (pp. 135–136), and [14] (pp. 425–430).

9　　Bredo C. Johnsen provides an account of how suspending belief in everything unclear might produce tranquility [5] (pp. 540-546). Brian Ribeiro proposes an alternative way to understand the connection between *epochē* and *ataraxia*, according to which the Pyrrhonist's relative tranquility is a result of his lack of beliefs about a range of issues, such as those in religion and ethics, that tend to weigh heavily on our minds [15] (pp. 327–329).

10　I have my doubts about how much comfort a skeptic would find in the thought that his torture may not truly be bad for him.

11　See, for example, [9], [16] (p. 278), [17] (p. 59), [18], [19], [20], and [21]. [22] provides a helpful summary of the most influential interpretations.

12　Robert J. Fogelin 1994: 5–9 also defends this interpretation and advocates for a contemporary skepticism of this sort [23] (pp. 5-9) . Describing such skepticism, he reports that it "leaves common beliefs, unpretentiously held, alone" [24] (p. 163).

13　Allowing the skeptic such value judgments may appear to be in tension with Sextus' claim that the skeptic achieves a relative peace of mind by suspending belief about what is truly good or bad. One possible way to resolve this apparent tension is to distinguish the everyday belief that something is good from the dogmatic belief that something is truly good, with the latter giving rise to more intensity. See, for example, [25] (pp. 75–80).

14　See, for example, [19] (p. 41) and [26] (pp. 117–118).

15　Tad Brennan and Casey Perin offer much more thorough defenses, although I disagree on some key details. See [25] (pp. 84–87 and 99–106) and [9] (pp. 7–32).

16　Of course, the Pyrrhonist will also continue to use the same techniques and continue to bring up the same types of considerations that have produced *epochē* in the past on such issues, and it may appear likely to him that these techniques and considerations will have similar results going forward. In fact, such a track record may leave the skeptic hesitant to accept some theoretical commitment even if he cannot yet produce an equally compelling argument against it [2] (1.33–34).

17　Moller, in fact, goes on to question whether *ataraxia* would appear valuable to the skeptic. Still, some goals must still appear valuable enough to motivate the skeptic to act, regardless of whether *ataraxia* is one of them.

18　See, also, [25] (p. 106).

19　The term "insulation" is taken from [27] (p. 115).

20　Johnsen offers this reading of Sextus as "at least plausible" [5] (p. 555).

21　This assumption strikes me as uncontroversial. If I am wrong about this, we can safely assume that a Pyrrhonist who considers the issue would suspend judgment on whether beliefs require concepts. As a consequence, his beliefs would continue to leave him open to the worry that he is using problematic concepts as he goes about his day.

22     For well-known defenses of such an interpretation, see [18] and [19].

23     While Perin discusses the propositional content of *phantasiai*, this should not lead us to assume that Sextus accepted the existence of propositions. On appearances and propositions, see [17] (pp. 14–16).

24     For a discussion of Galen's criticism of Pyrrhonism on this point and relevant citations, see [16] (pp. 146–147).

25     Mates makes this observation as well [17] (p. 20).

26     Theodore Sider, for instance, allows that we may be able to speak truthfully about tables and chairs in our daily lives even while arguing that, philosophically speaking, there are no tables and chairs [30].

27     See, for instance, [10].

28     Perhaps Sextus is guided on this issue by the common ancient assumption that we will not be emotionally invested in obtaining or avoiding something unless we believe that it is objectively good or bad. See [7] (p. 79n37) and [19] (pp. 45–46). As a sports fan, this assumption strikes me as implausible.

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
