# Peer review of "The Practical Price of Pyrrhonism"

_philosophies, doi:10.3390/philosophies8060104_

Round 1
Reviewer 1 Report
Comments and Suggestions for Authors
The paper argues that Pyrrhonian skeptics are subject to the worry that, in the living of their daily lives, they continue to hold dogmatic commitments. Or, put another way, to the worry that life as a true Pyrrhonist may not in fact be possible. The paper is well written and the author is clearly familiar both with the text of Sextus and with relevant secondary literature. Much of the paper consists of a presentation of the Pyrrhonist outlook among relatively standard lines; this is unobjectionable but not original. To the extent that there is novel argument, however, it does not strike me as convincing. The skeptic can be satisfied that daily life is possible simply by living it. There are some genuine issues about whether this will be the same kind of daily life as that lived by ordinary people, who may indeed hold certain views about the real nature of things, whether implicitly or explicitly. But this is not in itself a problem, and the skeptic does not need to engage in the kind of theorizing about the nature of concepts that is considered on pp.9-10. Indeed, the author agrees with this at the end of section 5 (which makes the point of the previous discussion somewhat unclear), but then finds this a source of worry, because "the skeptic would face considerable uncertainty about whether it is possible to live a relatively ordinary life without relying on potentially false dogmata" (p.10). I just don't see why this should be so; the skeptic simply goes ahead and lives life, while also engaging in philosophical discussion.
The question of what a "belief" is, and the question of what the skeptic's "investigation" consists in, are other issues that would deserve closer scrutiny than they receive in this paper.
Finally, a couple of small corrections: In the first sentence of section 5, "tell" should be "tells"; and the papers cited as Burnyeat 1998a and 1998b are in fact in a volume from 1997.
Author Response
Thank you for your thoughtful review. Please see the attachment for a description of the revisions and corrections I made based on your suggestions.

Reviewer 2 Report
Comments and Suggestions for Authors
This is an engaging and admirably clear discussion of the extent to which a Pyrrhonist might be able to live a tranquil, and somewhat ordinary, life. However, I think the author’s case rests entirely on an assumption that I find implausible, as I explain below.
If we understand the skeptic’s beliefs as more or less automatic responses to stimuli, it’s not clear that there are any concepts at work in such beliefs, contrary to the author’s assumption (8). For example, I don’t see why skeptically appropriate assent to the impression of a moving ship “must make use of some distinct conception of motion, a conception free from whatever problems appear to threaten its dogmatic counterparts” (9). It is unlikely that there is any conception of motion involved when a dog moves out of the way of a car, or when a small child moves to catch a ball rolling towards him; they have simply moved based on their experiences navigating the world. These may be understood simply as conditioned responses to stimuli. The idea that the skeptic is committed to some conception of motion seems to over-intellectualize a very ordinary event. When I observe an object in motion, I’m not aware of any conception of motion in general. I might reflect on this experience and think about what all instances of motion have in common, but whether I do so would have no effect on my ability to observe things in motion. If so, contrary to the author’s claim, the skeptic need not rely on concepts that are distinct from dogmatic ones, whether philosophical or ordinary. And if that’s right, the skeptic will in fact treat ordinary (or folk) theories of concepts as another dogmatic alternative (9).
Of course, the skeptic can think about concepts, but insofar as he need not rely on, assent to, or endorse any concepts in his life or philosophical practice, he need not be concerned about whether any of them are trustworthy or immune from doubt. So, in thinking about honey, the skeptic may avail himself of concepts articulated by philosophers or by ‘ordinary’ people, but he need not construct his own concept. This seems to be the clear lesson of Sextus’ response to the puzzle he raises at the beginning of PH 2 regarding how the skeptic can investigate dogmatic theories without apprehending them.
But perhaps I have not fully understood how and why the author thinks that even skeptically appropriate assent necessarily involves concepts.
Author Response
Thank you for your thoughtful review. Please see the attachment for a description of the revisions I made in response to your comments.

Round 2
Reviewer 2 Report
Comments and Suggestions for Authors
The revised paper adequately addresses my initial objection. And while I still disagree with the author’s position, I think the article is well worth publishing in its current form.
For what it’s worth, I would now complain in the following way. Suppose the skeptic suspends judgment on the controversy regarding the intentional content of the appearances that guide his action, both in day-to-day life and in philosophical investigation. If we were to accept Sextus’ claim that tranquility follows suspension, then presumably the skeptic would remain tranquil, despite being “unsure whether his appearances make use of problematic concepts” (12). I still don’t quite see why the skeptic would be (or should be) worried about the possibility that his ability to remain active relies on false dogmata as long as his uncertainty is an expression of his tranquil epochê. Further, if the skeptic understands his skepticism essentially as a practice or activity, then it seems to me at least, that issues regarding the philosophical justification or support for that activity need not disturb him. The simple fact that he remains active in ordinary life and in philosophy constitutes all the success he needs.